# *Albizia julibrissin* Exerts Anti-Obesity Effects by Inducing the Browning of 3T3L1 White Adipocytes

**DOI:** 10.3390/ijms241411496

**Published:** 2023-07-15

**Authors:** Yuna Kim, Hyanggi Ji, Dehun Ryu, Eunae Cho, Deokhoon Park, Eunsun Jung

**Affiliations:** Biospectrum Life Science Institute, Yongin 16827, Republic of Korea; bioyn@biospectrum.com (Y.K.); biocr@biospectrum.com (H.J.); biosc@biospectrum.com (D.R.); biozr@biospectrum.com (E.C.); pdh@biospectrum.com (D.P.)

**Keywords:** *Albizia julibrissin*, anti-obesity, 3T3L1, browning

## Abstract

This study investigated the effects of the *Albizia julibrissin* Leaf extracts (AJLE) on adipocytes using 3T3-L1 cells. AJLE inhibited adipogenesis by reducing the expression of peroxisome proliferator-activated receptor γ (PPARγ) and CCAAT/enhancer binding proteins (C/EBPs) that regulate enzymes involved in fat synthesis and storage, and subsequently reduced intracellular lipid droplets, glycerol-3-phosphate dehydrogenase (GPDH), and triglyceride (TG). AJLE also increased the expression of brown adipocyte markers, such as uncoupling protein-1 (UCP-1), PR/SET domain 16 (PRDM16), and bone morphogenetic protein 7 (BMP7) by inducing the differentiation of brown adipocytes, as shown by a decrease in the lipid droplet sizes and increasing mitochondrial mass. AJLE increased the expression of transcription factor A, mitochondrial (TFAM), mitochondrial DNA (mtDNA) copy number, and UCP-1 protein expression, all of which are key factors in regulating mitochondrial biogenesis. AJLE-induced browning was shown to be regulated by the coordination of AMPK, p38, and SIRT1 signaling pathways. The ability of AJLE to inhibit adipogenesis and induce brown adipocyte differentiation may help treat obesity and related diseases.

## 1. Introduction

The prevalence of obesity has reached epidemic proportions due to modern lifestyle, such as the Western diet and lack of physical activity [1,2]. Obesity is medically defined as body weight significantly higher than normal, with accompanying increases in the proportion of white adipose tissue (WAT). It plays a critical role in metabolic disorders-related diseases, such as diabetes, cardiovascular disease, hyperlipidemia, and hypertension [3]. Appetite suppressants (e.g., naltrexone) and lipolytic enzyme inhibitors (e.g., phentermine) have been developed as anti-obesity drugs but they have various side effects, including insomnia, nausea, constipation, and depression [4,5,6]. Thus, many attempts have been made to develop safe anti-obesity agents derived from natural products.

Adipocytes maintain lipid homeostasis and energy balance by storing fat in the form of triacylglycerol (TG) as an energy source and releasing free fatty acids in response to energy demand [4]. Fat mass increase in obese individuals is determined by adipocyte sizes and numbers [7]. The adipogenic process contributes to adipocyte size and number through different stages of growth arrest, mitotic clonal expansion (MCE), and differentiation. Members of the CCAAT/enhancer binding protein (C/EBP) transcription factor family, peroxidase proliferator-activated receptor γ (PPAR γ), and sterol regulatory element binding proteins (SREBPs) have been reported as key enhancers of adipocyte differentiation [8,9]. The expression of C/EBP β and C/EBP γ is associated with an early adipogenic differentiation program with conversion to MCE from preadipocytes. The activation of C/EBP α and PPARγ, which is a major transcription factor, play an important role in adipocyte differentiation, and their expression is mainly increased during the late differentiation process [10,11]. Inhibitors that suppress the differentiation of these preadipocytes have been proposed as potential anti-obesity treatments [12,13].

Adipose tissue can be divided into white, brown, and beige adipocytes, according to the functional and morphological characteristics of the adipocytes. White adipocytes can be converted to beige or brown adipocytes through thermogenic activity when exposed to a cold environment [14]. Unlike white adipocytes, brown adipocytes contain numerous mitochondria with multilocular lipid droplets and express high levels of uncoupling protein 1 (UCP-1), which regulate mitochondria biogenesis [15]. PR domain containing 16 (PRDM16) and bone morphogenetic protein 7 (BMP7) have been reported to up-regulate the level of UCP-1, and subsequently induce the browning process. PRDM16 induces brown adipocyte development in cooperation with peroxisome proliferator-activated receptor γ coactivator (PGC-1) α [16]. BMP7 increases the expression of UCP-1, PRDM16, and PGC-1α, through the p38 mitogen activated protein (MAP) kinase pathway [17]. In addition, the AMP-activated protein kinase (AMPK) and sirtuin 1 (sirt1) pathway contribute to the browning process by activating UCP-1. The compounds, which can stimulate the browning process, are considered as effective candidates for improving obesity status by increasing energy expenditure and decreasing fat accumulation [18].

*Albizia julibrissin*, which belongs to the family Leguminosae and is commonly known as the Persian silk tree or mimosa, is native to Japan, China, and Korea. It has been used to treat insomnia, amnesia, sore throat, and contusions. Flavonol and flavonoid glycoside from *A. julibrissin* have been reported to exert sedative and neuroprotective activity [19,20]. In addition, saponin from the stem bark of *A. julibrissin* showed anti-tumor and [21] anti-inflammatory activity [22]. However, the biological activity of *A. julibrissin* leaf extract (AJLE) on adipocytes remains to be elucidated.

The present study was performed to investigate whether AJLE exerted modulatory effects on 3T3-L1 adipocyte differentiation and the browning process.

## 2. Results

### 2.1. AJLE Inhibits Adipogenesis in 3T3-L1 Pre-Adipocytes

Cell viability was observed by using MTT and lactate dehydrogenase (LDH) assays to identify the cytotoxic concentration of AJLE. In the short-term cytotoxicity test, AJLE showed no significant change in cell viability up to a concentration of 50 μg/mL at 24 h and 48 h (Figure 1A). LDH was measured to confirm that the concentration selected from the short-term viability test did not exert cytotoxic effects in long-term treatment conditions. No cytotoxic effect of AJLE concentrations up to 50 μg/mL was observed on day 9 in differentiation conditions and day 15 (9 + 6) in browning conditions.

The inhibitory effect of AJLE on adipogenesis was confirmed by treating with AJLE (1, 10, 50 μg/mL) every 3 days for 9 days. On day 9, intracellular lipid droplets were increased with the differentiation of pre-adipocytes into adipocytes. As indicated by oil red O staining, AJLE significantly decreased lipid accumulation in a dose-dependent manner compared to the control group (Figure 1B). TG and glycerol-3-phosphate dehydrogenase (GPDH) enzymatic activity was observed to further characterize the effect of AJLE on adipocyte differentiation. GPDH is a key enzyme responsible for synthesizing TG in adipocytes [23]. As shown in Figure 1C, AJLE significantly inhibited GPDH enzymatic activity at 10 and 50 μg/mL compared to the control group. Consistent with the GPDH enzymatic activity results, AJLE decreased TG accumulation in a dose-dependent manner (Figure 1D).

To confirm the underlying regulatory mechanisms of AJLE on adipocyte differentiation, we observed the expression of PPARγ and C/EBPα, which are key regulators of adipogenesis. As shown in Figure 1E, AJLE significantly inhibited the expression of PPARγ and C/EBPα at 50 μg/mL compared with the control group. Taken together, AJLE inhibited adipocyte differentiation by suppressing PPARγ and C/EBPα expression.

### 2.2. AJLE Induces Brown Adipocyte-like Phenotype in 3T3-L1 Mature Adipocytes

To confirm the lipolytic effect, cells were treated with AJLE every 3 days for 6 days after the differentiation of pre-adipocytes into adipocytes for 9 days. The degree of lipolysis was confirmed by oil-red O staining. As shown in Figure 2A, there was no significant difference in lipid accumulation between the AJLE-treatment group and the control group. Interestingly, AJLE induced morphological changes in lipid droplets. Numerous small-sized lipid droplets, with morphology similar to that of differentiated brown adipocytes, were observed in the AJLE-treated group (Figure 2B, Appendix A). Figure 2C shows the general characteristics of white and brown adipocytes. The difference in lipid droplet morphology is one of the main factors determining adipocyte characteristics [15,24].

### 2.3. AJLE Increases Mitochondrial Function in 3T3L1 Mature Adipocytes

Based on these morphological characteristics, we hypothesized that AJLE could induce differentiation into brown adipocytes, which was confirmed by analyzing the mitochondria mass. Mitochondrial Transcription Factor A (TFAM) is a protein that regulates the transcription and replication of mitochondrial DNA and plays a crucial role in stimulating the function of mitochondria and cellular energy production. [25]. It also regulates the replication of mitochondrial DNA and maintains the appropriate quantity of mtDNA. As shown in Figure 3A,B, AJLE up-regulated the expression levels of TFAM in a dose-dependent manner. Since TFAM stimulates the replication of mitochondrial DNA, we observed the effect of AJLE on mtDNA copy number by analyzing the mtDNA/nDNA ratio (Figure 3C) and found that mitochondrial copy numbers were increased by AJLE. Moreover, an increase in mitochondria mass by AJLE was confirmed using MitoTracker Green FM immunofluorescence staining (Figure 3D) and flow cytometric analysis (Figure 3E).

### 2.4. AJLE Accelerates Browning-Related Gene Expression in 3T3-L1 Mature Adipocytes

We evaluated the expression of brown adipocyte marker genes to investigate the potential of AJLE to induce the conversion of white adipocytes to brown adipocytes. The results indicated that AJLE up-regulated the mRNA expression levels of key markers of brown adipocytes, such as UCP-1, PRDM16, and BMP7, in a dose-dependent manner (Figure 4A). The protein expression of PRDM16, PPARγ, and UCP-1 was also increased (Figure 4B). UCP-1, also known as thermogenin, is a mitochondrial transport protein found in brown adipocytes [15]. We observed that AJLE increased the expression of UCP-1 in mitochondria through immunofluorescence staining and co-localization analysis (Figure 4C).

### 2.5. AJLE-Induced Browning Is Regulated by the Coordination of AMPK, p38, and SIRT1/PGC-1 α Signaling Pathways

The AMPK, p38, and SIRT 1 pathway have been reported to contribute to the browning process by up-regulating UCP-1 [18,24]. Therefore, we observed the phosphorylation of AMPK and p38 in cells treated with AJLE. The expression level of p-AMPK and p-p38 was increased by AJLE (Figure 5A,B). Next, we examined the expression of PGC-1α and SIRT1. PGC-1α is a key transcription factor involved in regulating energy metabolism of brown adipocytes. SIRT1, known as a NAD^+^-dependent deacetylase enzyme, interacts with PGC-1α to modulate its function. SIRT1 deacetylates PGC-1α, increasing its activity [26]. The expression level of PGC-1α and SIRT1 was increased by AJLE. These results indicate that AJLE induced adipocyte browning by coordinating the AMPK, p38, and SIRT1/PGC-1 α signaling pathways (Figure 5C).

PGC-1α plays a crucial role in the mitochondria biogenesis of muscle cells [27]. We observed the effect of AJLE on the mitochondria biogenesis of muscle cells and its regulatory factors, such as PGC-1α and UCP-1. AJLE increased the expression of UCP-1 and PGC-1α and mitochondrial copy numbers in C2C12 cells (Appendix A). These results indicate that AJLE could contribute to energy expenditure in muscle tissue, which plays a significant role in metabolism and exercise-induced energy consumption.

### 2.6. High-Performance Liquid Chromatography Analysis

Phytochemicals, such as flavonoid and phenolic acid, have been reported to exert anti-obesity effects, including the browning process [18]. Previous investigations of the phytochemical component of *A. julibrissin* revealed that bioactive phenolic compounds, flavonoids, and cerebroside were contained in *A. julibrissin* extract [22,28].

We performed chemical profiling to identify the main component of AJLE. As presented in Figure 6, the high-performance liquid chromatography (HPLC) profile of AJLE showed several major peaks. The components were analyzed based on the retention times and UV spectra of standard phytochemicals. The standard phytochemicals for profiling analysis contain quercitrin (quercetin 3-O-rhamnoside), afzelin (kaempferol 3-O-rhamnoside), and hyperoside based on previous analytical reports on *A. julibrissin*. Previous studies reported the presence of quercetin derivatives from AJLE [29,30]. We identified these phytochemicals by performing a co-chromatographic analysis with reference chemicals for direct comparison. The HPLC analysis revealed that AJLE contained 1.49, 0.21, and 0.14% quercitrin, afzelin, and hyperoside, respectively.

## 3. Discussion

The present study explored the anti-obesity efficacy of AJLE in adipocytes and confirmed its regulatory role in adipocytes. AJLE both inhibited the differentiation of pre-adipocytes into mature adipocytes and converted white adipocytes into brown adipocytes. The regulation of mitochondrial biogenesis, lipid metabolism, and energy expenditure by inducing browning was also confirmed. Adipocytes store TG in the form of lipid droplets through the process of adipogenesis, and transcription factors, PPARγ and C/EBPs, act as major regulatory factors of this process [31,32]. When inducing differentiation in 3T3-L1 pre-adipocytes by AJLE treatment, the formation of lipid droplets was inhibited due to the reduced activity of GPDH, an enzyme that helps to synthesize TG (Figure 1B–D).

White and brown adipocytes show many morphological and functional differences [15]. White adipocytes form unilocular lipid droplets, have a small number of mitochondria, and store energy, whereas brown adipocytes form multilocular lipid droplets, have numerous mitochondria, and play a major role in energy expenditure (Figure 2C). Previous studies confirmed that 3T3-L1 cells could be differentiated into brown (beige) adipocytes by rosiglitazone (Rosi) and T3 treatment [33,34], these induced brown adipocytes were morphologically similar to AJLE-treated cells (Figure 2B). AJLE-induced browning was confirmed by the number and activity of mitochondria (Figure 3C–E) and the increased expression of UCP-1, a protein related to browning (Figure 4B,C). UCP1, also called thermogenin, is a protein that is concentrated in the inner mitochondrial membrane and releases energy as heat by uncoupling oxidative phosphorylation from ATP production. However, AJLE induced the browning of white adipocytes but failed to induce the differentiation of pre-adipocytes into brown adipocytes (Appendix A).

PRDM16 and PGC-1α are known to play important roles in determining pre-adipocyte fate and brown adipocyte development and function [35]. PGC-1α is considered a master regulator of mitochondrial biogenesis, one of the hallmark features of brown adipocytes [36]. Our results collectively demonstrate that AJLE stimulated a brown adipocyte-like phenotype in 3T3-L1 adipocytes. Obesity, marked by insulin resistance, is due to reduced BAT activation, followed by hyperlipidemia [37]. The combined effects of WAT activation and decrease in adipogenesis in activated BAT constitute an attractive and promising strategy for treating obesity and other metabolic complications. The present data clearly show that AJLE markedly activated brown adipocytes by enhancing the expression of key thermogenic markers, such as PRDM16, PGC-1α, and UCP1, at both the mRNA and protein levels (Figure 4A,B). Taken together, our results support the anti-obesity effect of AJLE through a selective thermogenesis program with BAT induction.

Recent studies revealed that the proliferation and apoptosis of brown adipocytes could affect the browning process. Apoptotic brown adipocytes enhance energy expenditure, and the gene influencing cell proliferation and apoptosis can regulate the browning of adipocytes [38,39]. In this study, AJLE up to a concentration of 50 μg/mL did not exert a cytotoxic effect on day 9 of differentiation condition and day 15 of browning condition. However, we cannot exclude the possibility that the change of cell proliferation or apoptosis by AJLE during long-term condition may affect the browning induction. To confirm whether the change in cell proliferation or apoptosis by the AJLE is involved in the browning process, further studies are needed.

AMPK is a key energy sensor molecule involved in energy homeostasis and metabolism. AMPK activation induced the fatty acid oxidation of long-chain fatty acids by activating acetyl-CoA carboxylase (ACC) enzymes, shifting the energy balance from storage to consumption [40], which confirmed that AJLE induced and activated the phosphorylation of AMPK, a signaling protein (Figure 6a). In addition, AJLE might induce the β-oxidation of fatty acids by up-regulating key mitochondrial proteins, such as TFAM. Based on this, we demonstrated that AJLE’s induction of brown adiposity could be mediated through the coordination of various signaling pathways (AMPK, SIRT1, and p-38) based on enhanced protein expression levels in AJLE-treated cells (Figure 6a–c). AMPK has also been reported to induce non-shivering thermogenesis through protein kinase A (PKA) [38].

The main peak in the phytochemical profiling analysis of AJLE was confirmed to be quercitrin, and the other two peaks were identified as afzeline and hyperoside. Previous studies suggested that quercitrin exerted an inhibitory effect on adipogenesis and stimulated lipolysis by regulating adipose triglyceride lipase (ATGL) and hormone-sensitive lipase (HSL) [41]. Afzelin was also reported to exert anti-obesity effects by inhibiting adipogenesis by modulating the AMPK signaling pathway [39]. Hyperoside exerted inhibitory effects on triglyceride accumulation and adipogenesis by regulating PPARγ, C/EBPβ, and SREBP-1c expression [42]. Based on those results, quercitrin, afzelin, and hyperoside can contribute to the anti-adipogenic and lipolytic effects of AJLE. However, no previous study has reported on the browning effect of AJLE-derived phytochemicals. Therefore, further studies are needed to confirm the effect of AJLE-derived phytochemicals on the induction of brown adipocytes.

In addition to brown adipocyte, muscle cells also play a critical role in energy expenditure through mitochondrial activation. In this study, we confirmed that AJLE increased the expression level of PGC-1α and mtDNA copy numbers in C2C12 cells (Appendix A). These results suggest that AJLE increases the activity of muscle cells through mitochondria biogenesis, which could contribute to a long-term positive effect on muscle degenerative diseases.

Taken together, AJLE could be used as a natural anti-obesity substance acting on multiple targets, including suppressing adipogenesis in adipocytes and stimulating energy expenditure in adipocytes and muscle cells.

## 4. Materials and Methods

### 4.1. Materials and Reagents

The 3T3-L1 cells (accession No. CL-173™) and C2C12 cells (accession No. CRL-1772™) were purchased from the American Type Culture Collection (ATCC; Manassas, VA, USA). Dulbecco’s modified Eagle’s medium (DMEM), fetal bovine serum (FBS), bovine calf serum (BCS), and penicillin-streptomycin (PS) were purchased from Welgene (Gyeongsan, Gyeongsangbuk-do, Republic of Korea). Insulin, dexamethasone, 3-isobutyl-1-methylxanthine (IBMX), formaldehyde, oil-red O, isopropanol, ethylenediaminetetraacetic acid (EDTA), dithiothreitol (DTT), rosiglitazone (Rosi), triiodothyronine (T3), and MTT (3-(4,5-Dimethylthiazol-2-yl) were purchased from Sigma-Aldrich (St. Louis, MO, USA). The TG assay kit was purchased from Cayman Chemical (Ann Arbor, MI, USA). The GPDH assay and DNA extraction kits were purchased from Takara Bio Inc. (Kusatsu, Shiga, Japan). The TRIzol, RNA-to-cDNA™ Kit, NuPAGE electrophoresis system, and horseradish peroxidase (HRP) chemiluminescent substrates (SuperSignal West Femo Substrate) were purchased from ThermoFisher Scientific (Waltham, MA, USA). Mito-Tracker Green FM was purchased from Molecular Probes (Eugene, OR, USA). AMPIGENE qPCR Green Mix Hi-ROX was purchased from Enzo Life Sciences Inc. (Farmingdale, NY, USA). PRO-PREP™ Protein Extraction Solution was purchased from iNtRON Biotechnology (Seong-Nam, Gyeonggi-do, Republic of Korea). PPARγ, C/EBPα, PRDM16, COX2, peroxisome proliferator-activated receptor gamma coactivator 1-alpha (PGC-1α), UCP1, UCP1 Alexa Fluor 647, and GAPDH antibodies were purchased from Santa Cruz Biotechnology, Inc. (Dallas, TX, USA). The p-AMPK, AMPK, p-p38, p38, and TFAM antibodies were purchased from Cell Signaling Technology (Beverly, MA, USA). SIRT1 antibody was purchased from LSBio (Lynnwood, WA, USA). Hyperoside (>98%) and quercitrin (>98%) were purchased from ChemFaces (Wuhan, China). Afzelin (99%) was acquired from Chirochem (Daejeon, Republic of Korea).

### 4.2. Preparation of Extracts from Albizia julibrissin Leaves

*Albizia julibrissin* leaves from a private farm (Jeju Island, Republic of Korea) were used for this study. Twenty grams of *A. julibrissin* leaves were extracted with 320 mL of water at 100 °C for 3 h. Then, the extract was cooled at room temperature and filtered. The aqueous extract was freeze-dried, resulting in a dark brown powder (2.79 g, yield 13.97%). Lyophilized powder was dissolved in dimethyl sulfoxide (DMSO) at a concentration 1000-fold higher than the final concentration in the medium.

### 4.3. Cell Culture and Differentiation

3T3-L1 pre-adipocytes were seeded in 6-well plates at a density of 1.5 × 10^5^ cells per well in DMEM containing 10% BCS until confluent. The cells were maintained at 37 °C in a humidified, 5% CO_2_ atmosphere. To differentiate the pre-adipocytes, 2 days after confluence, the cells were stimulated with DMEM containing 10% FBS, 0.5 mM IBMX, 1 μg/mL insulin, and 0.25 μM dexamethasone for 3 days (d). On d 3, the differentiation medium was replaced with 10% FBS/DMEM containing 1 μg/mL insulin. This medium was re-fed every 3 d. Pre-adipocyte and adipocyte were treated with vehicle or the indicated concentrations of AJLE every 3 d until the end of the experiment on d 9. Differentiation conditions and sample treatment methods are summarized in Appendix A. For differentiation to brown adipocytes, cells were additionally treated with 1 μM Rosi and 50 nM T3 for the indicated period according to the control protocol.

### 4.4. Cell Viability Assay

Cells were cultured in 48-well plates. After treatment for 24 or 48 h, MTT, a yellow tetrazole (5 mg/mL in phosphate-buffered saline, PBS) was added to the wells and incubated for 4 h at 37 °C. The supernatant was carefully removed, DMSO was added and mixed, and the absorbance was read at 563 nm.

### 4.5. Lactate Dehydrogenase Assay

The LDH assay was performed to determine the cytotoxicity by measuring LDH activity released from damaged cells. The assay was performed using an LDH Cytotoxicity WST Assay Kit (Enzo Life Sciences, Farmingdale, NY, USA) according to the manufacturer’s instructions.

### 4.6. Oil Red O Staining

Differentiated cells in 6-well plates were washed twice with PBS and fixed for 30 min with 10 % paraformaldehyde in PBS (pH 7.4). Cells were washed twice with distilled water, then stained for at least 1 h at RT in freshly diluted oil red O solution (six parts oil red O stock solution and four parts H_2_O, oil red O stock solution is 0.5 % oil red O in isopropanol). Oil red O-stained cells were dissolved with isopropanol, and absorbance was measured at 500 nm using a plate reader.

### 4.7. TG Assay

The cells were washed twice with cold PBS, and then harvested in 25 mM Tris buffer (pH 7.5) containing 1 mM EDTA. The harvested cells were sonicated three times for 15 s at 20% amplitude using UP50H with MS7 (Hielscher Ultrasonic GmbH, Teltow, Germany). TG was quantified using the TG assay kit, according to the instructions provided by the manufacturer.

### 4.8. GPDH Activity Assay

Cells were washed twice with cold PBS, and then harvested into 25 mM Tris buffer (pH 7.5) containing 1 mM EDTA and 1 mM DTT. The cells were homogenized by sonication after centrifugation at 12,000× *g* for 20 min at 4 °C, and GPDH activity was assayed in the supernatants using GPDH Assay Kit.

### 4.9. Western Blot Analysis

The cell lysates were prepared in PRO-PREP™ Protein Extraction Solution. Total protein was separated by the NuPAGE electrophoresis system (ThermoFisher Scientific) and transferred to polyvinylidene difluoride (PVDF) membranes. Immunoblotting was performed using primary antibodies against PPARγ, C/EBPα, PRDM16, COX2, PGC-1α, UCP1, TFAM, p-AMPK, AMPK, p-p38, p38, SIRT1, and GAPDH, and developed using goat anti-mouse IgG HRP and HRP chemiluminescent substrates. The amount of protein was quantified using ImageJ analysis software version 1.52a (National Institutes of Health, Bethesda, MD, USA).

### 4.10. qRT-PCR

Total RNA was extracted using TRIzol. The quantitation of RNA was performed using an Epoch microplate spectrophotometer (BioTek, Winooski, VT, USA). cDNA was synthesized using an amfiRivert cDNA Synthesis Platinum Master Mix. An AMPIGENE cDNA Synthesis Kit and an ABI7500 real-time PCR system (Ambion Inc., Austin, TX, USA) were used to perform qRT-PCR. Table 1 describes the primer sequences used in this study. Relative quantification of gene expression was normalized to that of GAPDH.

### 4.11. Mitochondrial Mass Analysis

The mitochondrial mass of adipocytes was analyzed using Mito-Tracker Green FM. Cells were harvested, resuspended in a PBS buffer containing 0.1% BSA (*w/v*), and then incubated with 500 nM MitoTracker Green FM in PBS buffer for 30 min at 37 °C. The cells were then centrifuged at 3000× *g* at 4 °C for 5 min and resuspended in 400 μL of PBS buffer. Fluorescence was analyzed using FACSCalibur.

### 4.12. Immunofluorescence

After differentiation of the cells in an 8-well slide chamber, the medium was removed, and the cells were washed with PBS, and fixed in 4% formaldehyde for 15 min at RT. After removing the fixative and washing with PBS for 5 min, the cell membranes were permeabilized with 0.1% triton x-100 for 10 min. The permeate was removed, washed with PBS for 5 min, blocked with 10% goat serum for 30 min at RT, and then overnight at 4 °C with a primary antibody (anti-UCP-1 (1:100)). The secondary antibody (anti-mouse IgG linked with Cy5 (1:500)) was incubated for 2 h at RT. For mitochondrial staining, cells were treated with 200 nM Mito-Tracker^®^ green for 1 h, and for nuclear staining, they were stained with 0.1 μg/mL of Hoechst 33,342 for 10 min. Analysis was performed using the EVOS^®^ FL Cell Imaging System.

### 4.13. mtDNA/nDNA Copy Numbere

The mtDNA copy number was analyzed using qPCR, as previously described [43]. Briefly, after differentiation, the cells were collected, and DNA was isolated from the cell. The mitochondrial-specific target gene selected for our assay was NADH dehydrogenase 1 (ND1), while the nuclear-specific target gene was hexokinase 2 (HK2). Table 2 describes the primer sequences used in this study. Analysis of the mtDNA/nDNA ratio can be calculated either by following the classical ΔΔCt method used for qPCR analysis or by calculating the number of mtDNA molecules per nDNA molecule. Alternatively, the number of copies of mtDNA can be calculated using the following formula:ΔCt = Ct(nDNA gene) − Ct(mtDNA gene)(1)
Copies of mtDNA = 2 × 2^ΔCt^(2)

### 4.14. HPLC Profile

HPLC analysis for identifying the phytochemicals in *Albizia julibrissin* leaf extract was performed using a Waters HPLC system (Waters, Milford, MA, USA) composed with Waters 2695 Separation module and a Waters 996 PDA. A Phenomenex Luna C18 column (4.6 × 250 mm^2^, ID. 5 μm) was used for separation, and the injection volume was 10 μL at 1.0 mL/min flow rate. A standard stock solution of quercitrin (47 µg/mL), hyperoside (6 µg/mL), and afzelin (10 µg/mL) was prepared for the injection. The *A. julibrissin* leaf extract was prepared at 5 mg/mL. The elution solvent consists of 0.1% trifluoroacetic acid in water (A) and acetonitrile (B). The mobile solvent gradient was as follows: 0–10 min, 90% solvent A; 10–20 min, 90% to 80% solvent A; 20–35 min, 80% to 70% solvent A; 35–45 min, 80% to 0% solvent A; and 45–55 min, 0% solvent A at RT. The detection wavelength was 254 nm.

### 4.15. Statistical Analyses

All experimental data are expressed as the mean ± standard deviation. Differences between the control and treatment groups were evaluated by the *t*-test, or analysis of variance (ANOVA), and performed using GraphPad Prism. Statistical significance was indicated as either *p* < 0.05 or *p* < 0.01.

## 5. Conclusions

Taken together, AJLE showed anti-obesity properties, including adipogenesis inhibitory effects through the PPARγ-C/EBPs pathway and brown adipocyte induction effect by increasing UCP-1, PRAM16, and BMP7 expression. In addition, AJLE-induced browning was regulated by the coordination of the AMPK, p38, and SIRT1 signaling pathways. AJLE increased mitochondrial biogenesis by up-regulating mtDNA copy numbers and TEAM and UCP-1 expression. These findings indicate that AJLE is a potential treatment for obesity and its associated metabolic disorders.

## Figures and Tables

**Figure 1 ijms-24-11496-f001:**
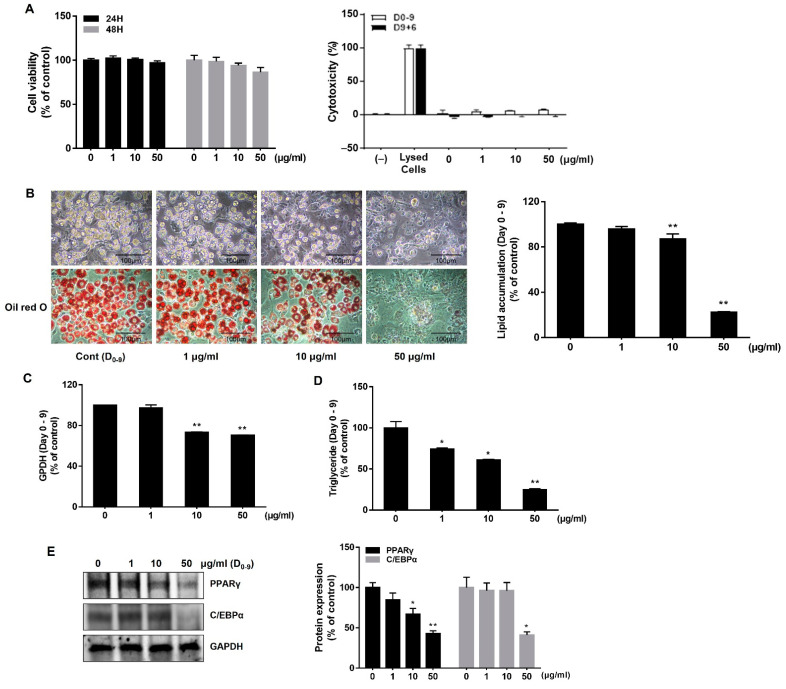
Effects of AJLE on adipogenic differentiation during the adipogenesis of 3T3-L1 pre-adipocytes. (**A**) Cell viability was confirmed by MTT assay at 24 and 48 h. Cytotoxicity in long-term conditions was confirmed by the LDH assay. (**B**) Pre-adipocytes were treated with AJLE for 9 d at the indicated concentrations to induce differentiation (medium was changed every 3 d), and then stained with oil-red O to observe lipid accumulation. The results confirmed adipogenesis inhibition by AJLE. (**C**) The cells were harvested, and GPDH activity in cells was analyzed. (**D**) The cells were harvested, and the intracellular TG amounts were measured. (**E**) The protein levels of adipogenic genes were confirmed by immunoblotting. The results are expressed relative to control cells after normalization to GAPDH. The results are expressed as the mean ± standard deviation (SD) (n = 3). * *p* < 0.05; ** *p* < 0.01, compared to control.

**Figure 2 ijms-24-11496-f002:**
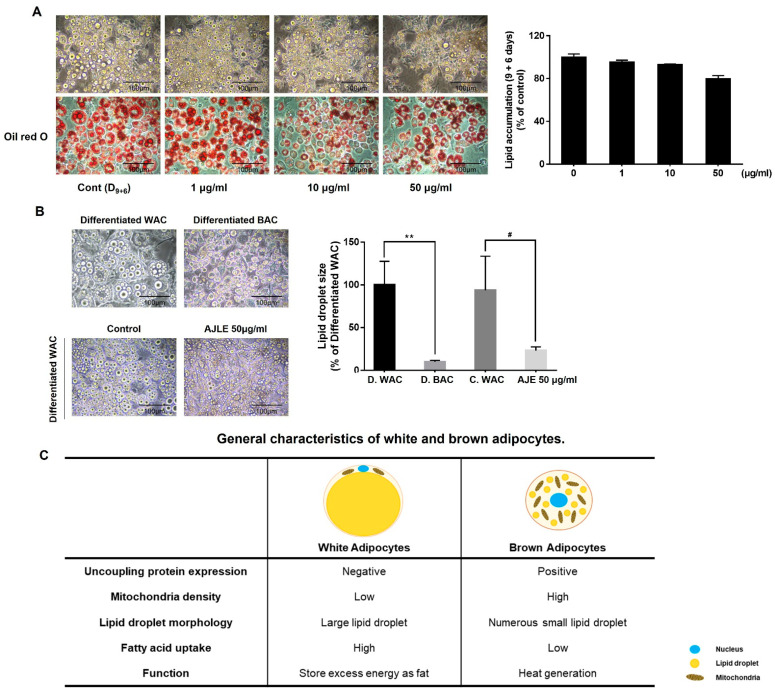
Morphologic changes in AJLE in 3T3-L1 mature adipocyte. (**A**) After the differentiation of pre-adipocytes into adipocytes for 9 d, the cells were treated with AJLE for 6 d (medium was changed on d 3) and lipid accumulation was measured by oil-red O staining. (**B**) Morphologic change by AJLE compared with differentiated brown adipocytes was confirmed, and lipid droplet size observed. (**C**) General characteristics of white and brown adipocytes. The results are expressed as the mean ± standard deviation (SD) (n = 3). ** *p* < 0.01, compared to D. WAC. # *p* < 0.05, compared to C. WAC.

**Figure 3 ijms-24-11496-f003:**
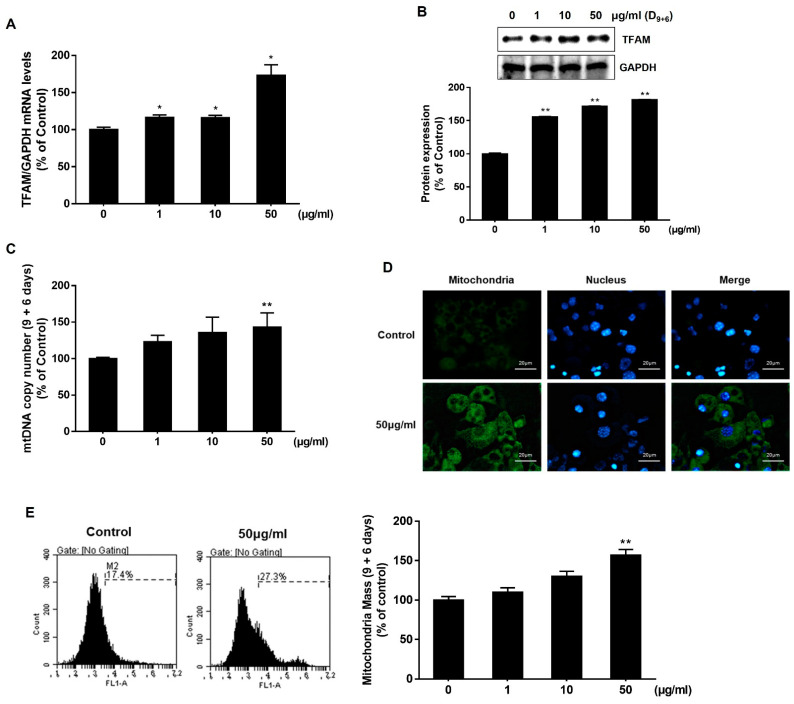
Analysis of mitochondrial mass in AJLE-treated 3T3-L1 mature adipocytes. Increased (**A**) TFAM mRNA and (**B**) protein in differentiated 3T3-L1 cells treated with the indicated concentrations of AJLE. Relative levels (%) of mRNA and proteins were normalized to GAPDH. (**C**) Analysis or calculation of the number of mtDNA molecules per nDNA molecules by AJLE. (**D**) Mitochondrial immunostaining and (**E**) flow cytometric analysis of fully differentiated 3T3-L1 cells treated with 50 μg/mL of AJLE. The results are expressed as the mean ± standard deviation (SD) (n = 3). * *p* < 0.05; ** *p* < 0.01, compared to control.

**Figure 4 ijms-24-11496-f004:**
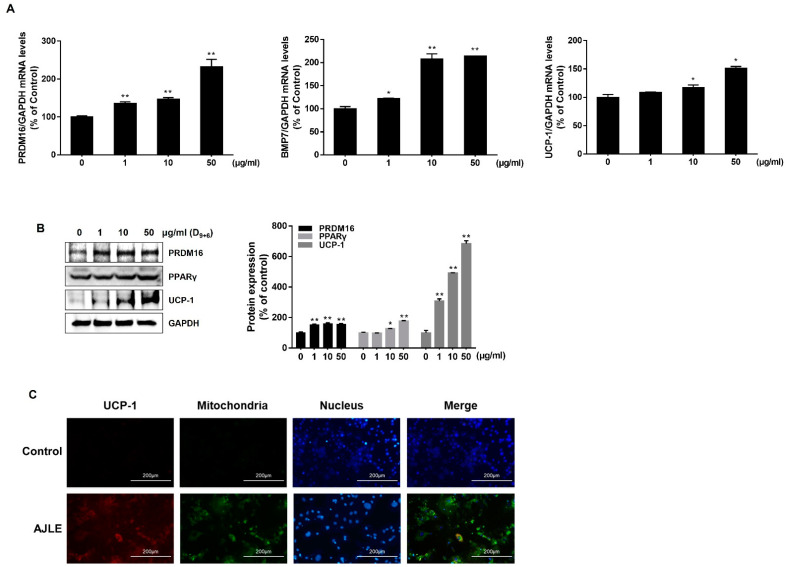
AJLE accelerates browning-related gene expression in 3T3-L1 mature adipocytes. (**A**) mRNA and (**B**) protein levels of brown adipocyte marker genes in differentiated 3T3-L1 cells treated with the indicated concentration of AJLE. (**C**) Immunostaining for UCP-1 (Red) and mitochondria (Green) of fully differentiated 3T3-L1 cells treated with the indicated concentrations of AJLE. The results are expressed as the mean ± standard deviation (SD) (n = 3). * *p* < 0.05; ** *p* < 0.01, compared to control.

**Figure 5 ijms-24-11496-f005:**
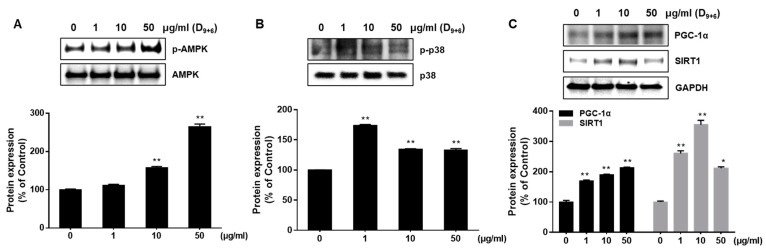
AJLE-induced browning is mediated by the coordination of AMPK, p38, and SIRT1 signaling pathways. Differentiated 3T3-L1 adipocytes were treated with AJLE at the indicated concentrations. The levels of (**A**) p-AMPK/AMPK, (**B**) p-p38/p38, (**C**) PGC-1α, and SIRT1 proteins were measured by Western blotting. The results are expressed as the mean ± standard deviation (SD) (n = 3). * *p* < 0.05; ** *p* < 0.01, compared to control.

**Figure 6 ijms-24-11496-f006:**
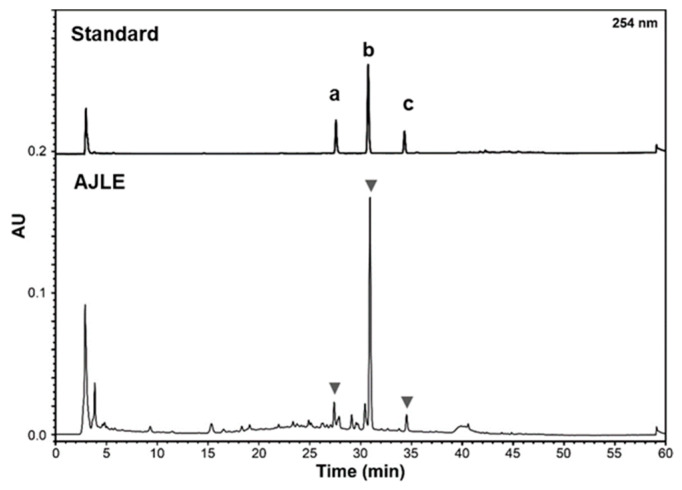
HPLC profile of AJLE. Standard peaks of hyperoside (a), quercitrin (b), and afzelin (c).

**Table 1 ijms-24-11496-t001:** Sequences of primers used qPCR.

Gene Name	Forward Primer	Reverse Primer
**PRDM16**	AAGACGTTCGGTCAGCTCTCCA	CTGGCACTCATGTGGCTTCTCT
**UCP-1**	AGATCTTCTCAGCCGGAGTT	AGCTGATTTGCCTCTGAATG
**BMP7**	GGAGCGATTTGACAACGAGACC	AGTGGTTGCTGGTGGCTGTGAT
**TFAM**	AATGTGGAGCGTGCTAAAAG	AGGGCTGCAATTTTCCTAAC
**GAPDH**	GACCCCTTCATTGACCTC	GCTAAGCAGTTGGTGGTG

**Table 2 ijms-24-11496-t002:** Sequences of primers used analysis of mtDNA/nDNA ratio.

Gene Name	Forward Primer	Reverse Primer
**ND1**	CTAGCAGAAACAAACCGGGC	CCGGCTGCGTATTCTACGTT
**HK2**	GCCAGCCTCTCCTGATTTTAGTGT	GGGAACACAAAAGACCTCTTCTGG

## Data Availability

The data presented in this study are available upon request from the corresponding author.

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
