# Peer review of "Albizia julibrissin Exerts Anti-Obesity Effects by Inducing the Browning of 3T3L1 White Adipocytes"

_ijms, 2023, doi:10.3390/ijms241411496_

Round 1

Reviewer 1 Report

Comments to Authors

In this manuscript, the authors eloquently describe how Albizia Julibrissin exerts anti-obesity activity through browning of white adipocytes. They demonstrate thorough knowledge of adipose tissue function as well as differences between white, brown and do a very nice job of describing the “beiging” process for the reader. I found this paper easy to read and follow. The authors demonstrate that AJLE inhibits pre-adipocyte differentiation into WAT, and also induces morphological changes of mature adipocytes resulting in “beiging”. The use of images to illustrate the changes in these cells was especially stunning. Authors measured markers that showed decrease in characteristics of WAT while increasing presence of BAT markers. It is obvious that this study was planned, and their conclusions are supported by the data and results presented. Overall, this was a very strong study and the manuscript is well done.

My only concern is the lack of n values in all of the figures. Error bars and significance is shown but n values are needed to be added.

General comments:

1.     In the results section 2.1 the authors describe proteins measured and reason for each but please include references.

2.     Line 113 “into a” could be better worded with the word “using”.

3.     Initially I was confused as to the purpose of (section 2.6) HPLC profile inclusion. Please include more information for this experiment in this section as you did with the other sections.

4.     Line 245 double space before AMPK.

5.     Line 315 there is an extra period at the end of the sentence.

6.     Line 378 there are two “triangles” before Ct method.

7.     I also think that including the muscle cell results didn’t add much but agree with putting it in the supplemental section.

Author Response

Reviewer 1

Thank you for your comments; they have helped us improve the quality of the manuscript. The comments are listed below with our responses. All changes were marked in red color.

Comments and Suggestions for Authors

In this manuscript, the authors eloquently describe how Albizia Julibrissin exerts anti-obesity activity through browning of white adipocytes. They demonstrate thorough knowledge of adipose tissue function as well as differences between white, brown and do a very nice job of describing the “beiging” process for the reader. I found this paper easy to read and follow. The authors demonstrate that AJLE inhibits pre-adipocyte differentiation into WAT, and also induces morphological changes of mature adipocytes resulting in “beiging”. The use of images to illustrate the changes in these cells was especially stunning. Authors measured markers that showed decrease in characteristics of WAT while increasing presence of BAT markers. It is obvious that this study was planned, and their conclusions are supported by the data and results presented. Overall, this was a very strong study and the manuscript is well done.

My only concern is the lack of n values in all of the figures. Error bars and significance is shown but n values are needed to be added.

--> We added n value in figure legend, as your comments.

General comments:

  1. In the results section 2.1 the authors describe proteins measured and reason for each but please include references.

--> We modified whole manuscript including results section 2.1, because one of reviewer pointed out problems with the English writing and result interpretation. We added the references, as your comments.

  1. Line 113 “into a” could be better worded with the word “using”.

--> We modified the Results paragraph 2.2 for the same reasons as above.

  1. Initially I was confused as to the purpose of (section 2.6) HPLC profile inclusion. Please include more information for this experiment in this section as you did with the other sections.

--> We added more information about HPLC result, as your suggestion

  1. Line 245 double space before AMPK.

--> We corrected the double space, as your suggestion

  1. Line 315 there is an extra period at the end of the sentence.

--> We corrected the sentence, as your suggestion

  1. Line 378 there are two “triangles” before Ct method.

--> We corrected the sentence, as your suggestion

  1. I also think that including the muscle cell results didn’t add much but agree with putting it in the supplemental section.

Reviewer 2 Report

In this study Kim et al. suggest that 1) Albizia Julibrissin leaf extract inhibits the differentiation of 3T3-L1 cells; and  that 2) Albizia Julibrissin leaf extract promotes the browning of 3T3-L1-derived adipocytes.

I consider that there are 3 critical issues with the present study:

1) There is insufficient evidence of the anti-adipogenic effect of Albizia Julibrissin leaf extract. In my opinion, the authors are underestimating the long-term toxicity of Albizia Julibrissin leaf extract, thus the cell viability assay should have been conducted at day 9. Also, a LDH assay could have been used to estimate the proportion of dead cells, effectively assessing cytotoxicity.

2) GAPDH participates in glycolysis, hence its RNA and protein expression is affected by substances modulating the energy-obtaining pathways of the cell. Therefore, this is not a good choice for reference gene in this study.

3) My main criticism is related to the browning effect of Albizia Julibrissin leaf extract. It is not possible to exclude the role of cell dead and de novo adipocyte differentiation in the present model. A brdu assay or similar test could give some insight into this hypothesis. Also, the authors should compare adipocytes treated with Albizia Julibrissin leaf extract for 6 days (9 + 6 days of culture) against adipocytes cultured for the same amount of time but not treated with Albizia Julibrissin leaf extract.

There are minor typos and grammar mistakes throughout the manuscript. As an example, in line 83, “To confirm the cytotoxicity of AJLE, it was treated for (24 and 48) h…”. This sentence is not grammatically sound because the subject of the action is not known. Also, I do not understand the purpose of parentheses in this sentence.

Author Response

Thank you for your comments; they have helped us improve the quality of the manuscript. The comments are listed below with our responses. All changes were marked in red color.

Comments and Suggestions for Authors

In this study Kim et al. suggest that 1) Albizia Julibrissin leaf extract inhibits the differentiation of 3T3-L1 cells; and that 2) Albizia Julibrissin leaf extract promotes the browning of 3T3-L1-derived adipocytes.

I consider that there are 3 critical issues with the present study:

1) There is insufficient evidence of the anti-adipogenic effect of Albizia Julibrissin leaf extract. In my opinion, the authors are underestimating the long-term toxicity of Albizia Julibrissin leaf extract, thus the cell viability assay should have been conducted at day 9. Also, a LDH assay could have been used to estimate the proportion of dead cells, effectively assessing cytotoxicity.

-->We agreed with your comment and added the data in result section. AJLE up to 50 μg/mL showed no cytotoxic effect on day 9 of differentiation condition and day 15 of browning condition.

2) GAPDH participates in glycolysis, hence its RNA and protein expression is affected by substances modulating the energy-obtaining pathways of the cell. Therefore, this is not a good choice for reference gene in this study.

-->Thank you for your valuable comment. GAPDH can participate glycolysis as your comment. However, there are many reports that GAPDH does not affect the expression of its related gene and the expression of GAPDH remains constant. (Ref.. 1.Vander Heiden MG, et al. Science. 2009 May;324(5930):1029-33. / 2. Wu N, et al. J Immunol. 2014 Aug;92(2):529-39. / 3. Stamatogiannopoulos G, et al. Cells. 2019 Nov;8(11):1410. / 4. Younesi V, et al. Wiley Interdiscip Rev Syst Biol Med. 2016 Nov;8(6):507-530). Also, in our study we confirmed that GAPDH showed no difference of expression. Therefore, we used it as a housekeeping gene.

3) My main criticism is related to the browning effect of Albizia Julibrissin leaf extract. It is not possible to exclude the role of cell dead and de novo adipocyte differentiation in the present model. A brdu assay or similar test could give some insight into this hypothesis. Also, the authors should compare adipocytes treated with Albizia Julibrissin leaf extract for 6 days (9 + 6 days of culture) against adipocytes cultured for the same amount of time but not treated with Albizia Julibrissin leaf extract.

--> We fully understand your concern. Based on cytotoxic result in above, AJLE up to 50 μg/mL did not exert cytotoxic effect in long-term condition. However, we cannot exclude the possibility that the change of cell viability or cell number during long-term condition may affect the result.

.

Comments on the Quality of English Language

There are minor typos and grammar mistakes throughout the manuscript. As an example, in line 83, “To confirm the cytotoxicity of AJLE, it was treated for (24 and 48) h…”. This sentence is not grammatically sound because the subject of the action is not known. Also, I do not understand the purpose of parentheses in this sentence.

-->We modified the manuscript as your suggestion.

Reviewer 3 Report

Major revisions

1)      The paper must be reviewed by a native English speaker.

2)      The Introduction section is too long and repetitive, providing many unrelated and superfluous information. Furthermore, the paragraph concludes with a summary of the findings. However,

since the introduction represents the section in which the Authors explain the assumption from which they started, it would be preferable to conclede the introduction with the aim of the work.

3)      In the Results paragraph 2.1, times and concentrations of the treatments carried out are reported, however there is no agreement between the two experiments, especially regarding the times considered. Moreover, superfluous information is reported regarding transcriptional factors, but this information is generally reported in other sections, such as introduction or discussion

4)      The paragraph 2.2 is poorly understandable for very bad English language, but also as the information is not reported with a suitable scientific language. It would be appropriate that important information, such as treatment times and drug concentrations, was reported in the paragraph and not in the caption of the figure.

5)      In the paragraph 2.3 Authors describe about TFAM (transcription factor A, mitochondria), but this factor is only named and its role is explained in no section. Since this is one of the key results, it would be useful for the Authors to dwell more on this concept.

6)      In the paragraph 2.6 Authors describe AJLE profile by HPLC. However, the explanation is too semplicistic and it should be described in deep. Indeed, it is not clear what is meant by “three major peaks” and what is the reason why this specific analysis was performed. Authors should clarify this aspect.

7)       In the discussion section the Authors should better explain the role of some proteins, such as SIRT1, which have been evaluated but whose role has not been well explained. Moreover, another cellular model was also used in the current study, C2C12, but the results obtained cannot be quickly described in the discussions, but should be better explained also in the results section.

Minor revisions

1)      Bibliography could be expanded

The paper must be reviewed by a native English speaker

Author Response

Thank you for your comments; they have helped us improve the quality of the manuscript. The comments are listed below with our responses. All changes were marked in red color.

Comments and Suggestions for Authors

Major revisions

1)      The paper must be reviewed by a native English speaker.

--> We modified whole manuscript as your suggestion.

2)      The Introduction section is too long and repetitive, providing many unrelated and superfluous information. Furthermore, the paragraph concludes with a summary of the findings. However, since the introduction represents the section in which the Authors explain the assumption from which they started, it would be preferable to conclede the introduction with the aim of the work.

-->Thank you for your valuable comments. We modified the introduction section as your suggestion.

3)      In the Results paragraph 2.1, times and concentrations of the treatments carried out are reported, however there is no agreement between the two experiments, especially regarding the times considered. Moreover, superfluous information is reported regarding transcriptional factors, but this information is generally reported in other sections, such as introduction or discussion.

--> We modified the Results paragraph 2.1 as your suggestion.

4)      The paragraph 2.2 is poorly understandable for very bad English language, but also as the information is not reported with a suitable scientific language. It would be appropriate that important information, such as treatment times and drug concentrations, was reported in the paragraph and not in the caption of the figure.

-->We modified the Results paragraph 2.2 as your suggestion.

5)      In the paragraph 2.3 Authors describe about TFAM (transcription factor A, mitochondria), but this factor is only named and its role is explained in no section. Since this is one of the key results, it would be useful for the Authors to dwell more on this concept.

-->We added the information about TFAM, as your suggestion.

6)      In the paragraph 2.6 Authors describe AJLE profile by HPLC. However, the explanation is too semplicistic and it should be described in deep. Indeed, it is not clear what is meant by “three major peaks” and what is the reason why this specific analysis was performed. Authors should clarify this aspect.

--> We added more information about HPLC result, as your suggestion

7)       In the discussion section the Authors should better explain the role of some proteins, such as SIRT1, which have been evaluated but whose role has not been well explained. Moreover, another cellular model was also used in the current study, C2C12, but the results obtained cannot be quickly described in the discussions, but should be better explained also in the results section.

--> We added more information about SIRT1 and C2C12 in result and discussion section, as your suggestion.

Minor revisions

1)      Bibliography could be expanded

--> We added the bibliography.

Round 2

Reviewer 2 Report

Dear authors,

Thank you for addressing my previous comments, which I consider to have been addressed adequately.

However, I consider that the authors' response to my previous comment "My main criticism is related to the browning effect of Albizia Julibrissin leaf extract. It is not possible to exclude the role of cell dead and de novo adipocyte differentiation in the present model.", should be adapted and included in the final manuscript. This discussion will be of interest to any reader of this paper, as it mention an alternative design for this study, and the reasons to adopt the present model.

Thank you

Author Response

Thank you for your comments; they have helped us improve the quality of the manuscript. The comments are listed below with our responses. All changes were marked in red color.

Comments and Suggestions for Authors

However, I consider that the authors' response to my previous comment "My main criticism is related to the browning effect of Albizia Julibrissin leaf extract. It is not possible to exclude the role of cell dead and de novo adipocyte differentiation in the present model.", should be adapted and included in the final manuscript. This discussion will be of interest to any reader of this paper, as it mention an alternative design for this study, and the reasons to adopt the present model.

-->We added the discussion, as your suggestion. [Line: 242-250]

Reviewer 3 Report

The Authors have addressed all of my concerns with the original manuscript

Only minor editing of English language are required

Author Response

Thank you for your comments; they have helped us improve the quality of the manuscript. The comments are listed below with our responses. All changes were marked in red color.

Comments and Suggestions for Authors

Only minor editing of English language are required.

-->Our manuscript was revised by Professional English editing services, as your suggestion.
